# Using persistent homology to understand dimensionality reduction in resting-state fMRI

## Abstract

Evaluating the success of a manifold learning method remains a challenging problem, especially for methods adapted to a specific application domain. The present work investigates shared geometric structure across different dimensionality reduction (DR) algorithms within the scope of neuroimaging applications. We examine reduced-dimension embeddings produced by a representative assay of dimension reductions for brain data ("brain representations") through the lens of persistent homology, making statistical claims about topological differences using a recent topological boostrap method. We cluster these methods based on their induced topologies, finding feature type and number — rather than reduction algorithm — as the main drivers of observed topological differences.

## 1 Introduction

The present work investigates shared geometric structure across different dimensionality reduction algorithms within the scope of neuroimaging applications. For most applications, a "dimensionality reduction" is any of a large class of methods that make inferences about structures underlying some data, typically to represent this data in both a more efficient and more interpretable way. Many dimensionality reduction (DR) problems can be equivalently formulated as "manifold learning" problems (i.e., estimating the manifold from which a dataset was sampled), and we will use the terms synonymously. Efforts to understand theoretical and empirical relationships between DR methods remain active[1−10].

The difficulty of relating dimensionality reductions can be compounded within specific application domains because methodologies often branch into variably specialized use cases. Nonetheless, specific use cases can also suggest more stringent criteria by which to compare dimension reduction outcomes. In functional neuroimaging, specialized dimension reduction algorithms proliferate the field, bridging disparate use cases, design philosophies, and biological motivations[11,12]. These DR algorithms share the goal of extracting networks of functional activity from resting-state fMRI brain data, and we will refer to them throughout as "brain representations." We compare brain representations in terms of the topologies they induce on a single set of shared data ("subject space") and the statistical robustness of the differences between them. We measure these topological statistics through persistent homology[13] and the related topological bootstrap[14,15].

**Problem Statement: Brain Representations & Subject Space**

Our primary goal is to compare the structural changes in a single neuroimaging dataset under a variety of brain representations. Because the structure of subject space in the original (un-reduced) data is unknown and impractical to compute, it is not feasible to grade brain representations' quality

34 by structure preservation. Instead, we group brain representations based on the similarity of the
35 subject-space structures they induce in reduced data.

36 We frame brain representation as a manifold learning problem, and we frame comparisons between
37 them as comparisons between estimated manifolds. We suppose our data lies on some manifold
38 $\mathcal{S} \hookrightarrow \mathbb{R}^D$ ("subject space"), from which we extract a finite dataset $\widehat{S}$ of $N$ samples. In this
39 work, $\mathcal{S}$ consists of resting-state fMRI scans, and $\widehat{S}$ is the Human Connectome Project (HCP)
40 Young Adult dataset; we then have $D \sim 10^8$ and $N \sim 10^3$. A **brain representation** is any
41 mapping $\widehat{\varphi}_i : \widehat{S} \to \mathbb{R}^{d_i}$ with $d_i \ll D$; for any brain representation, we define its corresponding
42 **induced subject space** $\widehat{S}_i = \widehat{\varphi}_i(\widehat{S})$. We compare brain representations $\widehat{\varphi}_i$ and $\widehat{\varphi}_j$ by comparing the
43 persistent homology of their induced subject spaces $\widehat{S}_i$ and $\widehat{S}_j$. To link the persistent homology and
44 manifold learning investigations, we make a key modeling assumption: there exists a local extension
45 $\varphi_i : \mathcal{S} \to \mathbb{R}^{d_i}$ of $\widehat{\varphi}_i$ that is a submersion in some neighborhood of $\widehat{S} \subset \mathcal{S}$. This assumption requires
46 that dimensionality reductions behave consistently on unseen data near training data, and constitutes
47 only a mild smoothness assumption on $\widehat{\varphi}_i$. Under this modeling assumption, we may assert the
48 existence of a manifold $\mathcal{S}_i \hookrightarrow \mathbb{R}^{d_i}$ containing $\widehat{S}_i$ such that the following diagram commutes:

$$
\begin{array}{ccc}
\widehat{S} & \xrightarrow{\ \widehat{\varphi}_i\ } & \widehat{S}_i \\
\big\downarrow & & \big\downarrow \\
\mathcal{S} & \xrightarrow{\ \varphi_i\ } & \mathcal{S}_i
\end{array}
$$

49 While the smoothness assumption on $\widehat{\varphi}_i$ is easily met by most DR algorithms, the connection between
50 the persistent homology of $\widehat{S}_i$ and the manifold $\mathcal{S}_i$ depends heavily on properties of the manifold
51 sampling $\widehat{S}$.

52 To compare brain representations $\varphi_i$ and $\varphi_j$, we compute dissimilarity metrics for all pairs of
53 points in $\widehat{S}_i$ and $\widehat{S}_j$ and examine the resulting Vietoris-Rips complex in each space. This approach
54 allows flexibility in the data and dissimilarities under consideration while still allowing claims about
55 DR-induced topological differences.

**Related Work: Persistent Homology & Dimensionality Reduction**

57 Most comparisons in the literature are primarily interested in grading the relative performance
58 of different DR algorithms. Though we do not share these goals, many comparative approaches
59 articulate frameworks and methods with important relationships to our own. We review a selection of
60 comparison methods, organized in roughly increasing order of the similarity between their goals and
61 framework to our own.

62 Some evaluation methods for dimensionality reduction lead with intuition, formalizing helpful
63 heuristics into rigorous ratings. We first reference Lee and Verleysen's co-ranking matrix[4], which
64 measures insertions to ("intrusion events") and deletions from ("extrusion events") $k$-neighborhoods
65 in the low-dimension vs. high-dimension space. While this measure is non-parametric (with respect to
66 the data geometry) and thus extremely flexible, it is sensitive only to local structure in the data. Later,
67 Lee and Verleysen showed that the performance of a DR method closely follows its (a) insensitivity
68 to norm concentration and (b) plasticity (i.e., the cost function gradient vanishing for distant points)[7],
69 leveraging a more geometric perspective to the analysis of DR performance than was typical of the
70 contemporary literature[9]. Extending this line of thinking, Wang et al[1] recently offered a strictly
71 empirical investigation of different DR methods in which they consider only the attractive and
72 repulsive forces of the loss function over varying distance scales. They show that this framework is
73 sufficient to characterize DR performance and extrapolate robust empirical principles of "good DR
74 methods" without need of an underlying formalism. While this work offers striking practical insights,
75 it does not offer an immediate path to describing degrees of divergence between DR methods.

76 We now consider a class of methods we characterize by their tendency to originate in formal,
77 geometric considerations of manifold learning. Singer and Wu's vector diffusion distance[8], an
78 extension of diffusion embeddings[17], uses local principal component analysis to locally estimate a
79 connection on the tangent bundle of a data manifold, from which it constructs a lower-dimensional

embedding. While the primary goal of their work is to define a manifold learning method, rather than a technique for comparing such methods, their vector diffusion distance explicitly captures a variety of geometric and topological data invariants. These invariants could be used analogously to our use of persistent homology to compare dimensionality reductions of the same dataset. Similarly, Tyagi et al[5] propose a set of tangent space estimation criteria for manifold sampling (as a function of manifold curvature and intrinsic dimension) that immediately suggest geometric routes of comparison between learned data manifolds. Finally, in more direct alignment with our goals, Sun and Machard propose a geometric theory of manifold learning, which comes equipped with an intrinsic metric[18,19]. Their approach is firmly grounded in classical information geometry[20,21], comparing learned models via the pullback of the Fisher information metric on direct probabilistic encodings of reduced data. While this perspective diverges substantially from our own, our constructions are mutually translatable, and their approach could provide interesting comparison and/or validation.

To the best of our knowledge, only two other studies[2,3] have examined dimensionality reduction through the lens of persistent homology. Both works primarily consider the recovery quality of a known manifold and propose quality metrics derived from persistent homology. Paul and Chalup[2] compare DR methods while varying manifold complexity, measuring performance by the similarity of pre- and post-DR Betti numbers as a function of sampling density. While their goals differ from ours, much of our comparison also hinges on counts of topological features; however, they did not have access to the topological bootstrap[14,15] we employ in our study. Rieck and Leitte[3] compute the Wasserstein distance between pre- and post-DR persistence diagrams as a metric of embedding quality. While their study is most similar to our own, there are several key differences. First, they operationalize persistent homology with a sublevel-set filtration of the local density function, whereas we use the Vietoris-Rips filtration on pairwise point dissimilarities. Second, they consider reduction of a surface embedded in $D = 3$ reduced to $d = 2$, whereas we consider data initially embedded in $D \sim 10^8$ and reduced to dimensionalities ranging from $d \sim 10^2$ to $d \sim 10^5$. Most importantly, we are able to leverage the topological bootstrap[14,15] in our work, which was not available at the time of their publication.

Finally, we also contrast our analysis of persistence data with a prevalent paradigm in persistent homology applications. Many persistent homology analyses[6,22−27] (including those above) operationalize the assumption that most topological information lives in a diagram's most persistent components, treating low-persistence generators as noise. Instead, we follow work in distributed persistence[28,29] and examine the distributional properties of our persistence diagrams to parse their topological content.

Our contributions in the present study are as follows: (1) A flexible framework for the statistical comparison of dimensionality reductions, applicable to any data and dissimilarity measure compatible with a Vietoris-Rips complex; (2) Robust statistical measurement of topological differences between dimension-reduced data; (3) Application to real neuroscience data over a diverse slice of widely used neuroimaging DR algorithms ("brain representation" or "BR").

## 2 Methods

### 2.1 Brain Data and Brain Representations

The data for this study consists of pre-processed resting-state functional MRI data from N=1003 Human Connectome Project young adult (HCP-YA)[30] subjects. Each subject's minimally pre-processed data consists of 91,282 spatial "grayordinates" by 1200 time points, giving an embedding dimension of $D \sim 10^8$ in the initial space. We then chose six different brain representations that are both common within the field and showcase the methodological variability of widely adopted techniques. The brain representations we consider can roughly be grouped by their underlying models of brain function. We characterize the first group of methods as seeking to cluster neural activity into spatially contiguous cortical "parcels." In the parcellation family, we have Yeo's parcellated networks[31], Glasser's multimodal parcellation[32], and Schaefer's local-global parcellation[33]. We also sample from a family of low-rank matrix factorization methods that parse non-contiguous networks of functional activity. Independent component analysis (ICA)[34], an extension and application refinement of PCA, underlies perhaps the most widely used brain representation in the field[35] and thus is represented here. In addition, we consider PROFUMO[36], which parses "functional modes" of brain activity from hierarchical Bayesian signal models. Finally, we include the "principal gradient"

134 (or "gradients")[37], a diffusion embedding method that organizes brain function through cortical
135 geometry.

136 From each brain representation, one or more feature types were computed to reflect the typical use of
137 brain representations in the neuroimaging literature. The five feature types considered in this work
138 are as follows: (1) "amplitude," the average power of the time signal in a given spatial component; (2)
139 "network matrix (netmat)," the matrix of pairwise Pearson similarities of time courses for each pair of
140 spatial components; (3) "partial correlation," the variance-normalized precision matrix; (4) "map,"
141 the spatial membership weights of a given spatial component in grayordinate space; and (5) "spatial
142 network matrix", the matrix of pairwise Pearson similarities of maps for each spatial component.
143 The decomposition rank, feature types, and number of features for each brain representation is
144 summarized in Table 1. Note that since subject data are encoded in terms of features, it is the feature
145 number and **not** the brain representation's decomposition rank that denotes the dimension $d$ of the
146 target embedding space in the mapping $\widehat{\varphi} : \widehat{S} \to \mathbb{R}^d$. We compare subject-space embeddings using
147 the pairwise dissimilarities of their points, which we compute as described in the next section.

| Representation Name | Decomposition Rank(s) $r$ | Considered Feature Type(s) | Feature Number(s) $d$ |
|---|---|---|---|
| PROFUMO | 33 | maps, spatial network matrices | $91282 \times 33, \binom{33}{2}$ |
| Dual-regression spatial ICA | 15, 25, 50, 100, 200, 300 | amplitudes, network matrices, partial network matrices | $r, \binom{r}{2}, \binom{r}{2}$ |
| Glasser parcellation | 360 | amplitudes, network matrices, partial network matrices | $360, \binom{360}{2}, \binom{360}{2}$ |
| Schaefer parcellation | 100, 200, 300, 600 | amplitudes, network matrices, partial network matrices | $r, \binom{r}{2}, \binom{r}{2}$ |
| Yeo parcellation | 17 | amplitudes, network matrices, partial network matrices | $17, \binom{17}{2}, \binom{17}{2}$ |
| Gradient (diffusion embedding) | 1, 15, 25, 50, 100, 200, 300 | maps | $91282 \times r$ |

Table 1: The combinations of brain representation, decomposition rank parameters, and feature types
investigated in the present work.

## 2.2 Dissimilarity Measures

149 For each brain representation method, decomposition rank within a given representation, and con-
150 sidered feature type, we compute pairwise distances between all subjects. Each feature type under
151 consideration is structured either as a vector (maps, amplitudes) or a symmetric positive semidefinite
152 (SPSD) matrix (network matrices). This bifurcation of data types is echoed in our choice of measures
153 when computing the dissimilarity between a pair of subjects. In both the vector case and the SPSD
154 data case, we ran our analysis using one dissimilarity measure intrinsic to the data type and another
155 derived from the Pearson correlation. We use Pearson-based dissimilarities in deference to the
156 ubiquitous use of the Pearson correlation in neuroimaging analyses.

157 We now define the dissimilarity measures we use on vector data. Suppose $s_i$ and $s_j$ are data vectors
158 in $\mathbb{R}^d$, and let $\rho(s_i, s_j)$ denote their Pearson correlation. Let $\langle \cdot, \cdot \rangle$ denote the usual inner product on
159 $\mathbb{R}^d$. We then define

$$d_{v_1}(s_i, s_j) = 1 - \langle s_i, s_j \rangle^2 \tag{1}$$

$$d_{v_2}(s_i, s_j) = 1 - \rho^2(s_i, s_j), \tag{2}$$

160 assuming the matrix $D_{ij} = \langle s_i, s_j \rangle$ is scaled to have entries in $[0, 1]$. Note that we can interpret $d_{v2}$
161 as approximately the angular distance between the vectors $s_i$ and $s_j$ after each has been centered. We
162 refer to $d_{v_1}$ as the "inner product divergence" and $d_{v_2}$ as the "Pearson divergence".

163 In the SPSD matrix case, we consider the geodesic distance between matrices on the Riemannian
164 SPD cone[38] alongside a (modified) Pearson divergence. The geodesic distance $d_{pd_1}$ on the symmetric
165 positive definite cone[39] is efficiently implemented via the approximate joint diagonalizer[40], and we
166 modify the Pearson divergence $d_{v_2}$ for the correlation matrix case by precomposing it with Fisher's
167 z-transformation[41] (the inverse hyperbolic tangent function): we write

$$d_{pd_2}(M_i, M_j) = \text{atanh}^* d_{v_2}(m_i, m_j), \tag{3}$$

where $m_i$ is the vector of upper-right triangle entries of the symmetric matrix $M_i$ (diagonal excluded). This precomposition is necessary for correlation matrices, as it normalizes the correlation values before re-correlating them. In contrast to the vector case, there is no simple comparison to be made between these two dissimilarity measures.

For each combination of brain representation, rank parameter, and feature type shown in Table 1, we compute pairwise dissimilarity according to both of whichever two measures are relevant. The subject-pairwise matrix of dissimilarities then forms the Gram matrix used to compute the persistent homology, as we describe in the next section.

## 2.3 Persistent Homology

We compute the Vietoris-Rips persistence[13] of each Gram matrix (which is obtained as described above), and we now give a very brief background on Vietoris-Rips persistence. For a thorough treatment of persistent homology, see Dey and Wang's text[13]; for a thorough treatment of algebraic topology preliminaries, see Hatcher's text[42].

### 2.3.1 Brief background

The topology of a space can be summarized by its *homology groups*, algebraic invariants that describe its structure. Persistent homology extends the constructions of homology to finite data, delivering a multiscale and threshold-free estimation of data topology. To compute the persistent homology of a dataset $X$, it must first be equipped with a *simplicial structure*: a simplicial complex $K(X)$ is a set of subsets of $X$ with the property that $\sigma' \in K$ whenever $\sigma' \subset \sigma$ for some $\sigma \in K$, and a *filtration* is a collection $\{K_t(X)\}$ such that $K_s \subset K_t$ when $s < t$. Homology groups $H_k(K_t(X))$ can be computed for each simplicial complex, and their *persistence* $\mathrm{PH}_k(X)$ is described by the evolution of these groups across the filtration. A simple example of a simplicial complex on $X$ is a graph $G(X)$. If that graph $G(X)$ is weighted, then the family $\{G_r(X)\}$ of graphs obtained from $G$ by thresholding its edges at weight $r$ is a filtration on $X$. If $G(X)$ is the graph on $X$ with edge weights given by the distance between vertices, then the filtration we just described is the *Vietoris-Rips filtration* on $X$. Given any dissimilarity matrix $d_X$, we can assume it is the Gram matrix of some graph $G(X)$ and compute its Vietoris-Rips persistence $\mathrm{PH}_k(X)$.

### 2.3.2 Topological bootstrap

Because it is possible (and, in fact, common) for multiple data elements to define the same homology generator, bootstrap re-sampling[43] is less straightforward in persistent homology than in many other modes of analysis. However, Reani and Bobrowski recently demonstrated a "topological bootstrap" method[14] that uses image persistence[44] to register homology generators found in co-embeddable spaces. If $X, Y$ can both be embedded into a shared space $Z$, then the inclusion maps $X \overset{\iota_X}{\hookrightarrow} Z$ and $Y \overset{\iota_Y}{\hookrightarrow} Z$ induce homology maps $\iota_X^*, \iota_Y^*$ with corresponding filtration maps $\iota_{r,X}^*, \iota_{r,Y}^*$ (assuming compatible filtrations on each space). A pair of nontrivial elements in $\mathrm{PH}_k(X)$ and $\mathrm{PH}_k(Y)$ is said to match via $Z$ if $\iota_{r,X}^*$ and $\iota_{r,Y}^*$ map them to the same nontrivial element of $\mathrm{PH}_k(Z)$ for some filtration value $r$. For a matched pair, the affinity score $\alpha$ of the match can be computed from ratios of lengths of intervals in each filtration for which elements in $\mathrm{PH}_k(X)$ and $\mathrm{PH}_k(Y)$ are matched via $Z$. We assign $\alpha = 0$ when no match is found and have $\alpha \in (0, 1]$ otherwise.

This procedure simplifies substantially in the bootstrapping case; we then have $Z = X$ and $Y = \widehat{X} \subset X$, and we need only check nontrivial elements of $\mathrm{PH}_k(X)$ for matches in $\mathrm{PH}_k(\widehat{X})$. In the bootstrap setting, Reani and Bobrowski measure the recurrence stability of a nontrivial generator $\eta \in \mathrm{PH}_k(X)$ by its *prevalence score*

$$\rho(\eta) \coloneqq \frac{1}{R} \sum_{j=1}^{R} \alpha(\eta, \widehat{\eta}_j), \tag{4}$$

where $\widehat{\eta}_j$ is the match of $\eta$ in the $j^{\text{th}}$ bootstrap. This is just the average affinity (over all bootstraps) between $\eta$ and its matches. In the present study, we compute prevalence scores for each generator in $\mathrm{PH}_1(X)$ for a given subject dissimilarity matrix $X$.

Our implementation[45] of the topological bootstrap is a mild extension of Garcia-Redondo et al's work[15], which efficiently integrates cycle registration with Ripser[46] and Ripser-image[47], refines the

cycle affinity measures proposed by Reani and Bobrowski, and broadens the conditions under which topological bootstrapping may be applied.

To satisfy the exchangeability criteria necessary for (any) bootstrapping, we also needed to account for family relationships between subjects in our bootstrap re-samples. Following the approach of Winkler et al.[48], we excluded all bootstrap re-samples that placed individuals with the same mother on different sides of the inclusion/exclusion divide. We conducted cycle registration using $R = 1000$ bootstraps per dataset at $90\%$ re-sampling (without replacement), and we consider $k = 1$-dimensional cycle registration in this work.

### 2.3.3 Prevalence-weighted Wasserstein-$p$ distance

The space of persistence diagrams is a metric space[49] under the Wasserstein-$p$ distance, which previous work[3] has used to compare persistence diagrams of different low-dimensional embeddings. To include statistical information about the stability of homology generators in this comparison, we define the *prevalence-weighted* Wasserstein-$p$ distance

$$W_p^{(\rho)}(d_1, d_2) := \left( \inf_{\gamma \in \Gamma_{12}} \sum_{x \in d_1} \|x \cdot \rho(x) - \gamma(x) \cdot \rho(\gamma(x))\|_\infty^p \right)^{\frac{1}{p}}. \tag{5}$$

Here, $d_1$ and $d_2$ are persistence diagrams, $\Gamma_{12}$ is the set of bijections between $d_1$ and $d_2$, and $\rho(x)$ is the prevalence of the homology generator $x$ given in 4. This is a simple re-weighting of the usual Wasserstein distance, modified to incorporate the prevalence score as a summary of per-cycle stability statistics.

### 2.3.4 The "matched Betti number" $\beta_k^{\text{(matched)}}$

We also define the "first matched Betti number" $\beta_k^{\text{(matched)}}$ as the number of matched cycles (i.e., matches with nonzero affinity scores) found in each bootstrapped re-sample. Intuitively, this is a count of the number of stable generators found in each bootstrap. The Betti numbers of a persistence module are typically summarized by curves, since each value of a filtration may induce a homology with a different set of Betti numbers. However, since the topological bootstrap already uses persistence interval information to find matched cycles and compute their affinity, we may consider $\beta_k^{\text{(matched)}}$ as having "collapsed" these curves via cycle registration. We consider the distribution of bootstrapped $\beta_k^{\text{(matched)}}$ values as a coarse summary of the distributed persistence[28,29] of a given dissimilarity matrix $d_X$.

### 2.4 Study Design

In Table 1, we lay out parameter and feature selections considered for each brain representation. For every representation, bootstrapped persistence is computed for all combinations of feature, parameter, and dissimilarity measure considered; this gives a total of 90 subject-pairwise dissimilarity matrices for which we compute $R = 1000$ topological bootstraps. We compute the prevalence-weighted Wasserstein-2 distance between all pairs of methods and the $\beta_1^{\text{(matched)}}$ distributions for each method. This method-pairwise distance matrix then undergoes Ward hierarchical clustering[50] to determine similarity. Our code is publicly available on github.

#### 2.4.1 Hypotheses

Comparing across feature and metric choices, we expect the SPD matrix geodesic distance to exhibit less sensitivity to concentration of measure and thus provide greater distinction between brain representations. We expect that within-feature groupings for map and amplitude will differ very little between the considered vector dissimilarity measures (equations 1 and 2). For all comparisons, we expect feature number and type to be more important drivers of differences than decomposition rank. Finally, within the PROFUMO analysis, we expect that spatial network matrices will be further from null than spatial maps, where we expect the very high dimensions of the spatial maps to suffer from concentration of measure.

Comparing across different brain representations, we expect to primarily see clustering according to (approximate) feature number and type, with secondary similarity clusters forming within each given

brain representation. We expect our analysis to align with previous results in the literature linking shared variance in brain representations[51−54], the details of which we expand upon in the results below.

# 3 Results

## 3.1 Persistent homology and dimension reduction

We first note several unexpected instances of trivial (or nearly trivial) persistence structure. First, full correlation matrices generated null $H_1$ persistence at every decomposition rank in every brain representation. By contrast, the partial correlation matrices (which is similar by conjugation to the inverse of the full correlation matrix) have interesting persistence for nearly all feature types, decomposition ranks, and dissimilarity measures. Additionally, the inner product divergence (1) generated trivial or almost trivial homology in both maps and amplitudes, across all ranks and representations; this is not true of the Pearson divergence, which we incorrectly hypothesized would exhibit similar behavior. A complete list of all methods that exhibited trivial $H_1$ persistence is given in Table S1.

### 3.1.1 Effect of embedding dimension

Our analysis saw that topological complexity (as measured by $H_1$ persistence) generally *decreased* with the number of features considered (Fig S1). Under the geodesic distance, mean prevalence score increased with feature number; for all other dissimilarity measures, mean prevalence score was not correlated with feature number (Fig S2). Taken together, these observations suggest that embeddings in higher dimensions elicit a smaller number of nontrivial $H_1$ generators which are also more robust. This runs counter to the consequences we might expect from concentration of measure in high dimensions, which pushes spaces towards the discrete topology (and thus a higher number of less stable generators). As expected, we also saw that feature number was a more important driver of persistence structure than the underlying rank of the decomposition (Fig S3).

### 3.1.2 Persistence vs. prevalence

We see evidence further corroborating Reani and Bobrowski's observation that the most prevalent cycles are not always the most persistent ones[14]. Figure 1 shows a sample persistence diagram in $H_1$ (colored by generator prevalence score) and a plot of all persistence-prevalence pairs observed in this experiment. Both plots demonstrate that cycles with low persistence can still have high prevalence, suggesting that the topological "noise" may carry meaningful structure in our data. In addition, we see a substantially richer difference structure between target embeddings when using the prevalence-weighted Wasserstein-2 distance instead of the classical Wasserstein-2 distance (Figure S4).

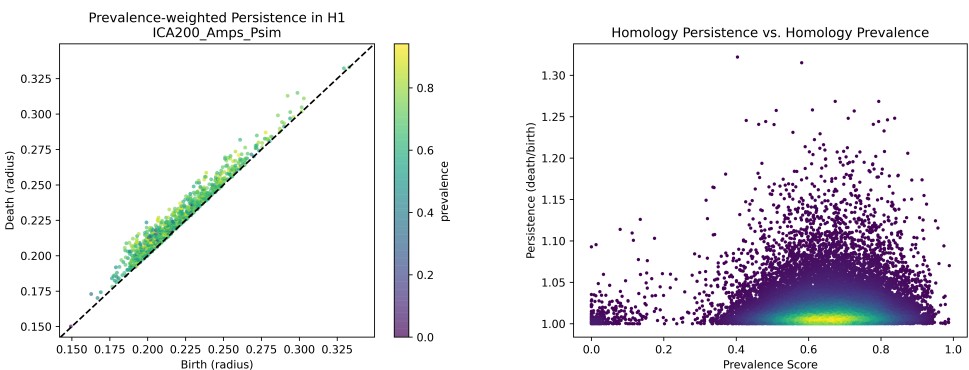

Figure 1: **(Left)** A sample persistence diagram, with color weights given by prevalence score. **(Right)** Persistence versus prevalence across all data collected, colored with a Gaussian kernel density estimator.

## 3.2 Persistence differences of brain representations

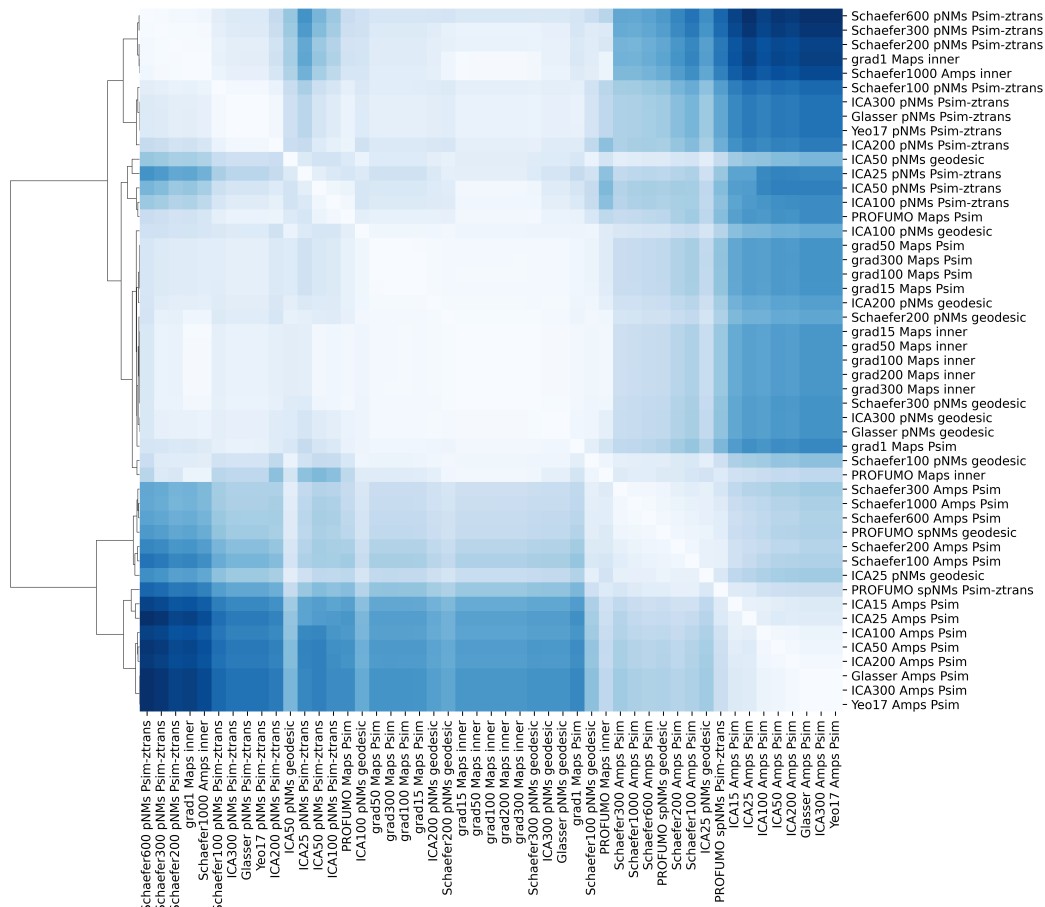

Figure 2: Prevalence-weighted 2-Wasserstien distances between $H_1$ persistence diagrams for all pairs of methods combinations with nontrivial first homology. Ward hierarchical clustering gives the dendrogram on the left side of the plot, which organizes labels into groups that maximally share variance. Lighter colors denote smaller distances, while darker (blue) colors denote larger ones.

The prevalence-weighted Wasserstein distance makes its strongest distinction between amplitudes and network matrix/spatial map feature types, which form the two main diagonal blocks and highest dendrogram branches (Fig 2). As hypothesized, this implies that our method distinguishes more strongly between feature type (and number) than between brain representation type, which forms the next set of blocks and branches. This is still somewhat surprising, however, because brain representations differ substantially in terms of whether they are unilateral or bilateral, binary or weighted, and decomposition rank.

We are also surprised to see PROFUMO spatial network matrices in the amplitude block. Both amplitudes[55] and spatial network matrices[12] have been shown to be highly sensitive to individual differences in behavior, but these feature types are interpreted very differently. Amplitudes may be linked to within-network synchronization[56], within-network plasticity[57], or within-network interneuron function[58], whereas spatial network matrices are indicative of between-network shared brain regions that may play a role in cross-network integration[12]. Both amplitudes and spatial netmats have higher test-retest reliability (i.e., within-subject stability) than the features in the other block[36,59]. Given this context, the clustered blocks of the prevalence-weighted Wasserstein may constitute a segregation of trait-sensitive (amplitude and spatial network matrix) from state-sensitive (temporal network matrix) features. This observation highlights the need for an evaluation method that can detect *which* elements of the persistence module are shared across representations, rather than only being able to similar topologies of subject similarity.

## 3.3 Computational resources

All other computations, including cycle registration, were negligible in cost compared to the computation of persistence modules for all bootstraps — roughly $270,000$ persistence modules were computed in total. Memory demands remained relatively low ($\leq 50$GB per homology computation). Our implementation was embarrassingly parallel on a queue-managed HPC cluster. We estimate that this experiment used approximately $80,000$ CPU hours over the course of a month. Computation of image-persistence was the most costly individual step, with each embedded persistence module taking 1-3 hours to compute (compared to order of 10 minutes or less for other persistence computations).

# 4 Discussion

## 4.1 Conclusions

Our method reveals interesting relationships between dimensionality reductions of resting-state fMRI data. The prevalence-weighted Wasserstein distance distinguishes much more strongly between feature type than dimensionality reduction, potentially segregating trait-sensitive from state-sensitive features. Notably, this distinction holds without regard to choice of dissimilarity measure.

Without exception, full network matrices gave rise to trivial $\mathrm{PH}_1$ modules. Persistence modules generated from the inner product divergence (1) were (approximately) trivial as well, in sharp contrast to those generated from the Pearson divergence (2); this suggests that amplitude and spatial map features of brain representations tend to be "mean-dominated," in the sense that per-subject deviations from group-level structures are typically small.

In addition, we saw a counterintuitive decrease in persistence "complexity" as a function of increasing embedding dimension, which highlights the difficulties of evaluating dimension reduction in high-dimensional target spaces. We also examined the relationship between persistence and prevalence, finding that the two are largely uncorrelated for our data. Coupled with the stronger distinctions realized by the prevalence-weighted Wasserstein-2 distance, we believe that persistence and prevalence may be somewhat complementary as measures of cycle importance.

## 4.2 Limitations

Because of the high cost of parameter exploration, dimensionality reduction computation, and topological bootstrapping, only a few dimensionality reduction methods were examined in this work. An extension of this analysis to a wider array of brain representations may be warranted, especially newer methods that derive an explicitly geometric basis for functional activity (e.g., Laplacian eigenvalues[60]).

Another important limitation of our work is the very high dimension-to-sample-size ratio ($N \ll d$) of our data. In this regime, it is difficult to ascertain what features we see because of structure in the data and what topological features are products of the curse of dimensionality. This could be partially ameliorated by conducting our analysis over adequately constructed null data and comparing the results, which is beyond the scope of this work.

## 4.3 Future Directions

In addition to addressing some of the limitations noted above, we offer several directions for follow-up work on this study. First, we propose a consideration of the per-bootstrap Wasserstein distance between methods; a distributional picture of differences in the endogenous metric of persistence modules could yield important insights. Second, it is possible to repurpose the topological bootstrap to track the addition/deletion of homology components by different brain representation; practically, this is primarily hindered by the lack of a suitable dissimilarity metric between pairs of points under different embeddings. Finding and validating such a metric would be a valuable direction of inquiry. Finally, we wish to suggest an investigation into the theoretical properties of the prevalence-weighted Wasserstein metric.

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
