# Supplementary Materials

# Using persistent homology to understand dimensionality reduction in resting-state fMRI

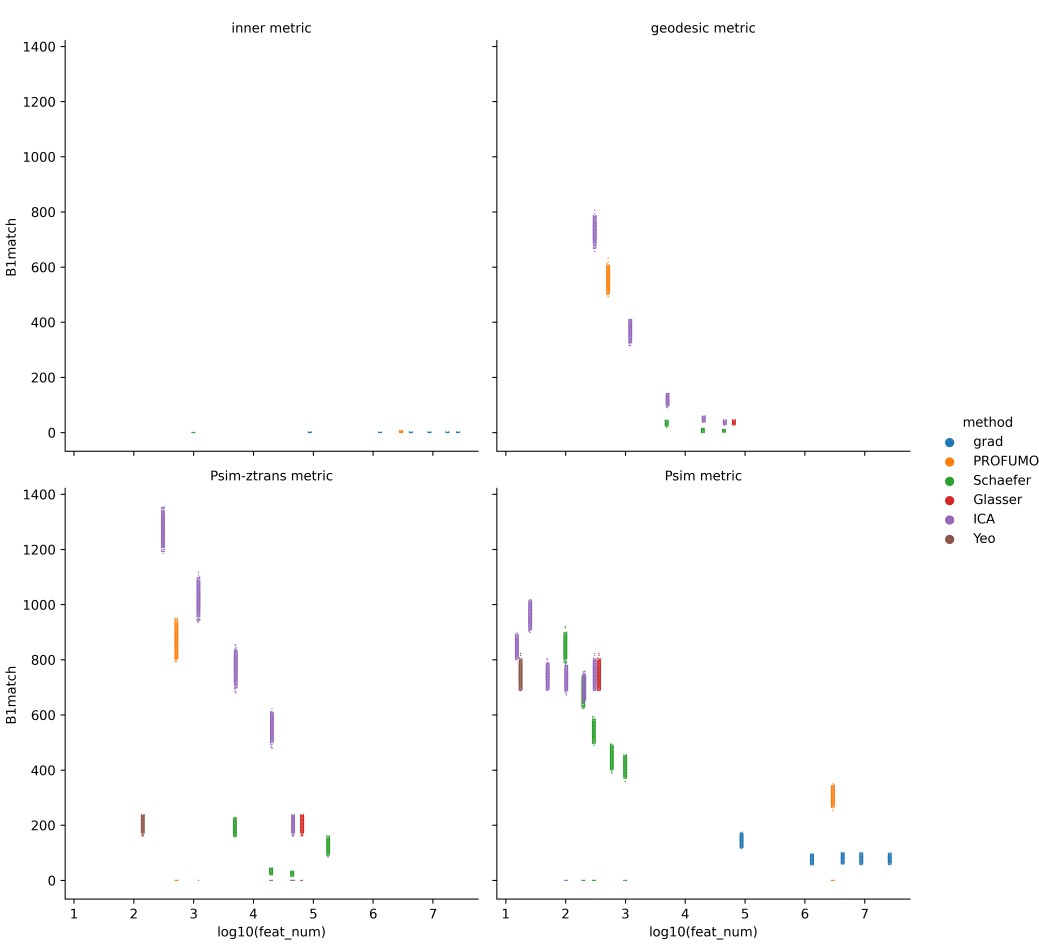

Figure S1: Number of matched cycles as a function of feature number, separated by brain representation type, feature, and dissimilarity metric. PROFUMO features are spatial netmats for geodesic and $z$-transformed Pearson measures and maps otherwise. Diffusion gradient features are maps. All other features are either partial netmats (under geodesic or $z$-transformed Pearson measures) or amplitudes (under Pearson or inner product divergence).

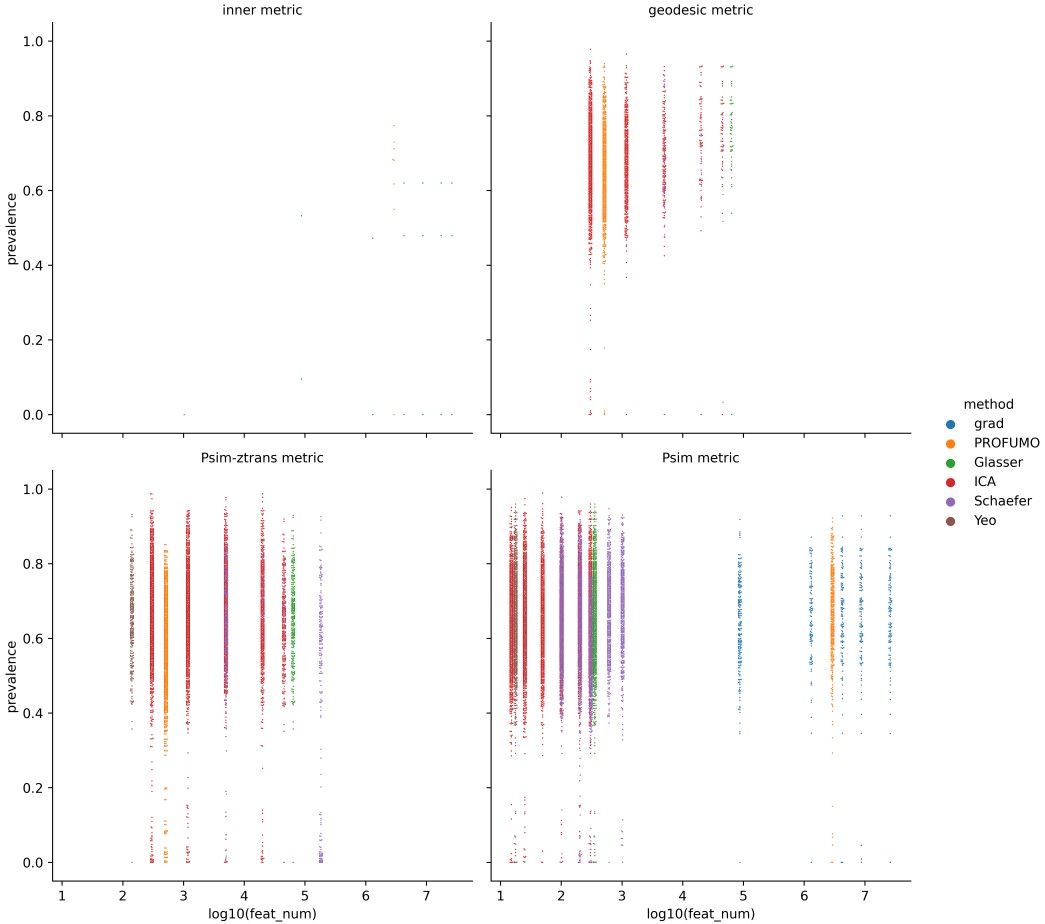

Figure S2: The distribution of prevalence scores (average match affinity over bootstraps) does not vary with feature number, except under the geodesic distance. For partial and spatial network matrices under the geodesic distance, we see that generator sparsity and generator stability both ***increase*** as a function of feature number.

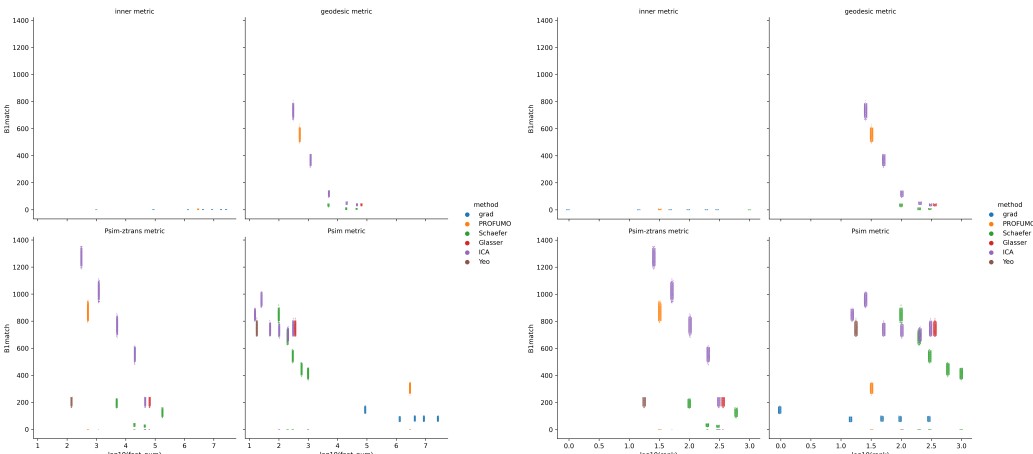

Figure S3: The distribution of numbers of matched cycles, as a function of both feature number **(Left)/Figure S1** and decomposition rank **(Right)**. We observe approximate monoticity as a function of feature number, but not as a function of decomposition rank, suggesting that feature number is a more important driver of persistence structure than decomposition rank.

Wasserstein Dist. from
H1 persistence diagrams

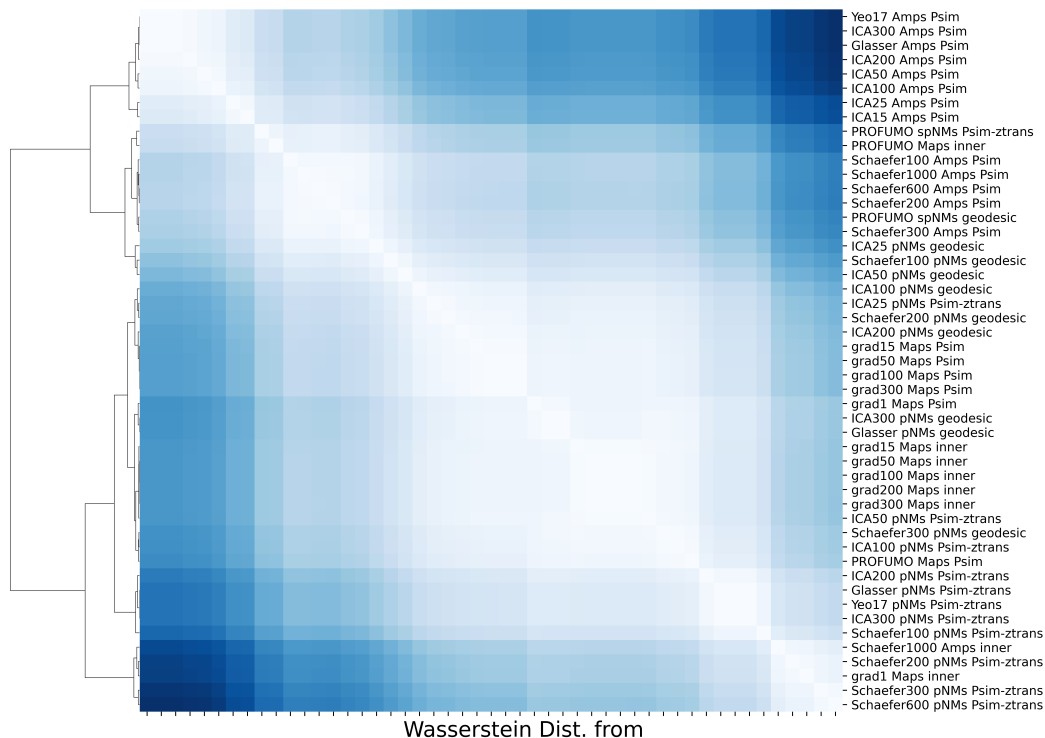

Wasserstein Dist. from
H1 persistence diagrams
(prevalence-weighted)

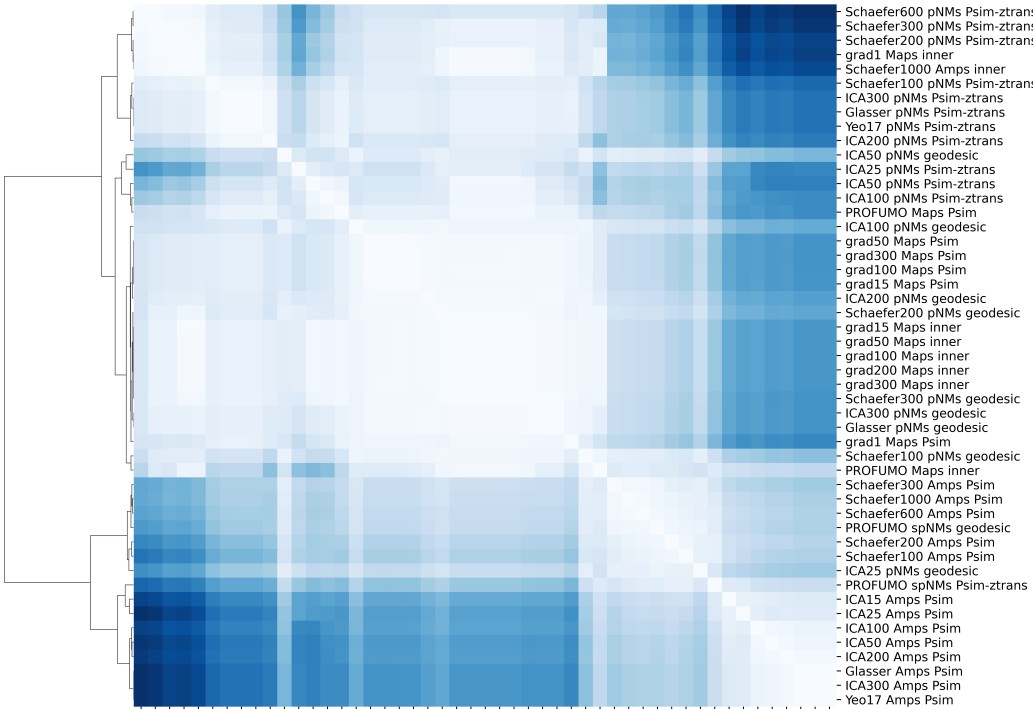

Figure S4: Clustermaps between all methods from the Wasserstein-2 distance **(Left)** and its prevalence-weighted variant **(Right)**. The prevalence-weighted distance shows greater differentiation between methods.