# OpenReview forum: "Using persistent homology to understand dimensionality reduction in resting-state fMRI"
_NeurIPS.cc/2023/Conference — Submitted to NeurIPS 2023_

### Official Review · Reviewer_MybT · 2023-07-03

**Soundness:** 2 fair
**Presentation:** 2 fair
**Contribution:** 1 poor
**Rating:** 3
**Confidence:** 3

**Summary:**

The paper is an empirical study of different dimensionality reduction (DR) methods for brain activity data.
Authors compare various approaches to brain representation using the MRI data from Human Connectome Project.
In the manuscript, "brain representations" are called "dimensionality reduction" (DR) since they present brain MRI data in a compact way.
By DR, the author mean ways of presenting brain activity: selecting brain segmentation (parcellations), measuring activity inside these zones, calculating cross-correlation etc. The definition is different from the one used in ML/AI community, where by DR we mean algorithms like t-SNE, UMAP, etc.
Authors use topological data analysis (persistent homology, topological bootstrap, prevalence score) to evaluate the quality of DR.
Lots of computational resources were (80, 000 CPU hours over the course of a month) were spent to such an evaluation.

**Strengths:**

Neuroimaging is an active field of research. The paper is technically correct in my opinion. Some recent tools of TDA (like prevalence, topological bootstrap) are applied.



**Weaknesses:**

First of all, I'm not an expert in neuroscience, so I can evaluate only the dimensionality reduction/topology part.

1. In the manuscript, "brain representations" are called "dimensionality reduction" (DR) since they present brain MRI data in a compact way. By DR, the authors mean ways of presenting brain activity: selecting brain segmentation (parcellations), measuring activity inside these zones, calculating cross-correlation etc. The definition is different from the one used in ML/AI community, where by DR we mean algorithms like t-SNE, UMAP, etc.
2. No contribution for ML/AI/DL. Typical conclusion from the study: "As expected, we also saw that feature number was a more important driver of persistence structure than the underlying rank of the decomposition". The conclusion gives insights about types of brain features, not DR algorithms from ML.
3. I didn't understand some parts of the paper with neuroscience jargon (cortical parcels, grayordinates, subject space). Captions on Fig. 2 are not explained, what does mean "Shaefer600 pNMs Psim-ztrans", etc?
4. Some references are missing and some relevant methods are not evaluated.

Overall, the paper seems to fit more traditional neuroscience community than NeurIPS community.
But I can be mistaken.

**Questions:**

1. Why not to evaluate representations after DR via Representation Topology Divergence [1]?
2. Why you don't use standard tools for DR, like t-SNE, UMAP?
3. Recently, some tools for topologically-aware dimensionality reduction were proposed [2, 3].
Can they by applied for your problem?
4. The topological bootstrap and prevalence are rather novel tools in topological data analysis. Given significant computational budget (bootstraping and calculating persistence diagrams 1000 times) are there real benefit in using it?

[1] Barannikov, S., Trofimov, I., Balabin, N., & Burnaev, E. (2021). Representation topology divergence: A method for comparing neural network representations. arXiv preprint arXiv:2201.00058.

[2] Moor, M., Horn, M., Rieck, B., & Borgwardt, K. (2020, November). Topological autoencoders. In International conference on machine learning (pp. 7045-7054). PMLR.

[3] Trofimov, I., Cherniavskii, D., Tulchinskii, E., Balabin, N., Burnaev, E., & Barannikov, S. (2023). Learning topology-preserving data representations. ICLR' 2023.

**Limitations:**

Authors adequately addressed the limitations.

---

> ### Author Rebuttal · Authors · 2023-08-10
>
> **Weaknesses:**
>
> 1. [...] The definition is different from the one used in ML/AI community, where by DR we mean algorithms like t-SNE, UMAP, etc.
>
>     * We apologize for our confusing terminology. As the reviewer correctly notes, we focus on domain-specific features sets that typically involve a brain parcellation and feature choice (see the ‘Terminology’ section in general response). Although these specific feature sets differ from generalizable DR approaches, we believe that our flexible mathematical framework has broader applications for DR comparisons (see ‘Relevance to NeurIps’ section in general response).
>
> 2. [...] The conclusion gives insights about types of brain features, not DR algorithms from ML.
>
>     * We believe that the flexible mathematical framework developed here can be used across domains and for domain-general DR comparisons (see ‘Relevance to NeurIps’ section in general response).
>
> 3. Captions on Fig. 2 are not explained, what does mean "Shaefer600 pNMs Psim-ztrans", etc?
>
>     * We apologize for this oversight. In this example, "Schaefer" refers to the parcellation choice (the Schaefer parcellation), "600" refers to the rank of that parcellation, pNMs is shorthand for "partial network matrices," and "Psim-ztrans" denotes that the Fisher-transformed Pearson divergence was used as the dissimilarity measure for this embedding.
>
> 4. Some references are missing and some relevant methods are not evaluated.
>
>     * We apologize for this oversight.
>
> Overall, the paper seems to fit more traditional neuroscience community than NeurIPS community. But I can be mistaken.
> * We believe our general mathematical framework, which encompasses comparisons of embeddings/networks/DRs/etc, is of interest to researchers in the NeurIps community (see ‘Relevance to NeurIps’ section in general response).
>
> **Questions:**
>
> 1. Why not to evaluate representations after DR via Representation Topology Divergence [x]?
>
>     * We were not aware of this method and agree that it is very interesting and shares many goals with our analysis. We are excited to validate against it in future work. However, we believe that our method is applicable to a larger class of representation comparisons since it does not require a shared vertex set and may provide more granular accounting of the degree of difference between representations.
>
> 2. Why you don't use standard tools for DR, like t-SNE, UMAP?
>
>     * One of our primary goals for this work was to address the challenge of analytical flexibility within the neuroimaging domain (see the ‘Novelty and contributions’ section in our general response), and we therefore chose domain-specific feature sets. Nevertheless, we believe our flexible mathematical framework has domain-general applications (see ‘Relevance to NeurIps’ section in general response).
>
> 3. Recently, some tools for topologically-aware dimensionality reduction were proposed [x, x]. Can they by applied for your problem?
>
>     * Our goal was to evaluate the impact of analytical flexibility that is typical to large sectors of the neuroimaging community; as such, these fall outside of our scope. However, these methods do fall within the purview of our comparison framework and look extremely interesting; it is possible that we will be able to include them in future work, and we appreciate the reviewer bringing them to our attention.
>
> 4. The topological bootstrap and prevalence are rather novel tools in topological data analysis. Given significant computational budget (bootstraping and calculating persistence diagrams 1000 times) are there real benefit in using it?
>
>     * We apologize for not motivating this decision more clecomparison arly in the manuscript. Our persistence data tends to primarily lie near the birth-death line (see **Figure 1** in the general response figure page); however, some work has found important structure in this topological “noise” [e.g., 1]. However, most techniques make statistical claims on persistence intervals, rather than homology generators themselves [2-4]. Our data do not exhibit high-persistence topological features, but do show nontrivial topological structure. The inclusion of the topological bootstrap offers insight to data that exhibits topological structure other than a small number of very persistent features, extending the reach and sensitivity of our comparison framework.
>
> [1] P. Bubenik, M. Hull, D. Patel, and B. Whittle, “Persistent homology detects curvature,” Inverse Problems, vol. 36, no. 2, p. 025008, Jan. 2020, doi: 10.1088/1361-6420/ab4ac0.
>
> [2] B. T. Fasy, F. Lecci, A. Rinaldo, L. Wasserman, S. Balakrishnan, and A. Singh, “Confidence sets for persistence diagrams,” Annals of Statistics, vol. 42, no. 6, pp. 2301–2339, Mar. 2013, doi: 10.1214/14-AOS1252.
>
> [3] R. J. Adler, S. Agami, and P. Pranav, “Modeling and replicating statistical topology, and evidence for CMB non-homogeneity,” Proc. Natl. Acad. Sci. U.S.A., vol. 114, no. 45, pp. 11878–11883, Nov. 2017, doi: 10.1073/pnas.1706885114.
>
> [4] E. Onaran, O. Bobrowski, and R. J. Adler, “Functional Central Limit Theorems for Local Statistics of Spatial Birth-Death Processes in the Thermodynamic Regime.” arXiv, Feb. 23, 2022. doi: 10.48550/arXiv.2202.02766.

---

> > ### Comment · Area_Chair_bQox · 2023-08-18
> >
> > Dear reviewer,
> >
> > Thanks for your support during the review process. The authors have provided additional clarifications to your questions—please acknowledge the rebuttal briefly and ask any additional questions you might have.
> >
> > Thanks,\
> > Your AC

---

> > ### Comment · Reviewer_MybT · 2023-08-19
> > **Response**
> >
> > I appreciate authors for addressing my questions. In my opinion, the manuscript has potential and novelty (applications of prevalence scores, study of DR methods for neuroimaging data). But the text itself should be improved by taking into account comments from my review and other reviews. I don't see an updated version of the manuscript here. So, I prefer to leave my evaluation unchanged.

---

### Official Review · Reviewer_af6F · 2023-07-04

**Soundness:** 3 good
**Presentation:** 1 poor
**Contribution:** 2 fair
**Rating:** 3
**Confidence:** 4

**Summary:**

This paper investigates shared geometric structure across different (very broadly speaking) dimension reduction algorithms for functional brain connectivity. The authors examine different connectivity representations through persistent homology via topological statistical and bootstrap.

**Strengths:**

- The paper is written clear language.
- The paper proposes novel metrics to evaluate graph structures.

**Weaknesses:**

- This paper is quite ambiguously written and not self-contained although the languages are clear.
- I believe the presentation can be far improved by explaining background better, restructuring paragraphs and better description of mathematical notations.
- For example, topological sampling is not well explained and readers would have to rely on other papers. Also, sections do not flow smoothly as mathematical notations are either not consistent or variables in equations do not connect.
- There is no baseline experiments and the result from the analysis cannot be properly validated.


**Questions:**

- The authors mention that the primary goal is to compare the structural changes in a single neuroimaging dataset under a variety of brain representations, which is not so clear. Where is the "changes" coming from?
- Where is $D \sim 10^8$ in line 40 coming from? This is explained far later in line 123 and very data specific.
- $d$ in section 1 is dimension, then it becomes dissimilarity in section 2.2, and then it becomes persistent diagrams in section 2.3.3.
- There is no baseline or comparisons with other methods. At least there should some naive approach that should demonstrate the benefits of the proposed analysis.


**Limitations:**

There is a section describing the limitation.

---

> ### Author Rebuttal · Authors · 2023-08-10
>
> **Weaknesses:**
>
> This paper is quite ambiguously written and not self-contained although the languages are clear.
>
> * We apologize for our lack of clarity. Please see the ‘Terminology’ section in the general response for further information.
>
> I believe the presentation can be far improved by explaining background better, restructuring paragraphs and better description of mathematical notations.
>
> * Thank you for this feedback, which we will make sure to implement in our future work.
>
> For example, topological sampling is not well explained and readers would have to rely on other papers. Also, sections do not flow smoothly as mathematical notations are either not consistent or variables in equations do not connect.
>
> * We apologize for inconsistencies in our mathematical notations, and we regret that we cannot give a detailed treatment of the topological bootstrap in the space provided.
>
> There is no baseline experiments and the result from the analysis cannot be properly validated.
>
> * We have performed some baseline experiments using an alternative domain-specific method (‘canonical correlation analysis’)  -- see **Figure 2** of the figure page in the general response. Unfortunately, we were unable to include this work due to space restrictions, but this will be published in our future work.
>
>
> **Questions:**
>
> The authors mention that the primary goal is to compare the structural changes in a single neuroimaging dataset under a variety of brain representations, which is not so clear. Where is the "changes" coming from?
>
> * We apologize for our lack of clarity. When discussion ‘changes’ we meant to refer to differences in subject embeddings due to analytical flexibility.
>
> Where is D~10^8 in line 40 coming from? This is explained far later in line 123 and very data specific.
>
> * This is the dimensionality of the original acquired neuroimaging data. Although we agree that this number is data-specific, we note that the general problem of high dimensional data analysis is encountered in many different domains.
>
> d in section 1 is dimension, then it becomes dissimilarity in section 2.2, and then it becomes persistent diagrams in section 2.3.3.
>
> * We apologize for the inconsistencies in our mathematical notations.
>
> There is no baseline or comparisons with other methods. At least there should be some naive approach that should demonstrate the benefits of the proposed analysis.
>
> * We have performed some baseline experiments using an alternative domain-specific method (‘canonical correlation analysis’)  -- see **Figure 2** of the figure page in the general response. Unfortunately, we were unable to include this work due to space restrictions, but this will be published in our future work.

---

> > ### Comment · Reviewer_af6F · 2023-08-17
> >
> > I appreciate the authors for their comments and clarifications. However, I agree with SXyw on several weaknesses of this paper, e.g., lack of clarity, motivation and self-containedness, as well as lack of baseline experiments, which should have been there in the original manuscript. Venues like NeurIPS is not a journal paper that reviewers can keep their eyes on until the manuscript gets revised until publication, therefore, I would have to maintain my current score.

---

### Official Review · Reviewer_Qnet · 2023-07-07

**Soundness:** 2 fair
**Presentation:** 1 poor
**Contribution:** 2 fair
**Rating:** 3
**Confidence:** 3

**Summary:**

The authors study shared geometric structure across different dimensionality reduction (DR) algorithms applied to neuroimaging data (fMRI data from the Human Connectome Project). In particular, they compare different DR algorithms (which they call "brain representations") by applying them to the same data sample and comparing the resulting Vietoris-Rips complex in each low rank data embedding using a modified topological bootstrap and cluster on the resulting estimated topologies.

**Strengths:**

The authors do introduce framework for the comparison of DR methods that work with any data or dissimilarity measure amenable to Vietoris-Rips filtration and apply their method to real neuroscience data.

**Weaknesses:**

The authors seem to have put in a good amount of work. However the paper currently lacks motivation or clear significance. Instead the paper reads like a number of different parcellations and DR methods were all applied to a public dataset after which a number of somewhat justified, somewhat arbitrary sequential analysis decisions were made to arrive at a clustering to determine similarity. Good background is given on each step, but it is not clear where the novelty lies here, or why these steps are the right ones. It is unclear exactly what deeply useful conclusions can be drawn from this analysis.

The writing and definitions could be much more clear throughout. Some are used before they are defined well, some terms with precise meaning seem misused (e.g., induced topology seem misused; "induced on...data"), some terms are highly redundant and misleading (e.g., "brain representations").

The github link is not really anonymized (there is another project at the same github link clearly from the "Personomics Lab").

**Questions:**

"Brain representations" is confusing, sounds like something it isn't, and results in redundant confusing phrases like "We frame brain representation as a manifold learning problem" when you've already said you would use the terms "brain representation", "dimensionality reduction" and "manifold learning" synonymously. Why not just use "DR method" as you do in lines 62-112?
After wading through this relatively confusing exposition, I am disappointed to be left with little idea as to the impact of the findings. Evidently methods cluster according to feature number and type...why does this warrant publication in NeurIPS? What is the major new insight into DR methods enabled by this approach?

**Limitations:**

The authors have a limitations section.

---

> ### Author Rebuttal · Authors · 2023-08-10
>
> The writing and definitions could be much more clear throughout. Some are used before they are defined well, some terms with precise meaning seem misused (e.g., induced topology seem misused; "induced on...data"), some terms are highly redundant and misleading (e.g., "brain representations").
>
> * We apologize for the lack of clarity in our definitions and any imprecise uses of terminology in our manuscript. For the specific example given: we intended this phrase in the sense of a "metric-induced topology," since we are only able to realize projection topologies via some choice of dissimilarity measure. Since not all of our measures of dissimilarity are actually metric, we settled on "induced topology" as an (hopefully evocative) alternative to "metric-induced topology" in the manuscript. However, the reviewer is obviously correct to point out that this is unlikely to coincide with the subspace topology on our embeddings (considered as embedded submanifolds of the "original" subject space S), and we are happy to shift our terminology to avoid this misimpression in the future.
>
> The github link is not really anonymized (there is another project at the same github link clearly from the "Personomics Lab").
>
> * We apologize for this oversight.
>
> **Questions:**
>
> "Brain representations" is confusing, sounds like something it isn't, and results in redundant confusing phrases like "We frame brain representation as a manifold learning problem" when you've already said you would use the terms "brain representation", "dimensionality reduction" and "manifold learning" synonymously. Why not just use "DR method" as you do in lines 62-112?
>
> * We apologize for the lack of clarity in our definitions. Please see the section ‘terminology’ in the general response for more information.
>
> [...] why does this warrant publication in NeurIPS? What is the major new insight into DR methods enabled by this approach?
>
> * We believe that our method addresses important domain-specific challenges of analytical flexibility (see ‘Novelty and contribution’ section in general response) and provides a flexible mathematical framework that has future extensions for broader domain-general comparisons of dimensionality reduction approaches (see the ‘Relevance to NeurIps’ section in general response).

---

> > ### Comment · Area_Chair_bQox · 2023-08-18
> >
> > Dear reviewer,
> >
> > Thanks for your support during the review process. The authors have provided additional clarifications to your questions—please acknowledge the rebuttal briefly and ask any additional questions you might have.
> >
> > Thanks,\
> > Your AC

---

### Official Review · Reviewer_SXyw · 2023-07-07

**Soundness:** 3 good
**Presentation:** 3 good
**Contribution:** 1 poor
**Rating:** 3
**Confidence:** 3

**Summary:**

The paper proposes an approach of comparison of different standard dimensionality reduction technics (DRT) of fMRI, using topological data analysis tools. In this case, these DRT computes lower dimensional representations of the Human Connectome Project dataset.
Then, for this specific dataset, the authors associate to each DRT (and for each feature type of each DRT) divergence matrices computed using correlations of these representations.
Finally, these matrices, seen here as distance matrices, are compared to each other, by
 1. Computing persistence diagrams associated to the Vietoris-Rips complex of these matrices,
 2. Reweighting these diagrams by *prevalence scores*,
 3. Computing wasserstein distances between these diagrams,
 4. Cluster the DRTs using this distance matrix.

**Strengths:**

Looking at the fundamental differences between different dimension reduction techniques is a very interesting topic: on one hand, this can help increasing the performance of these methods by cleverly taking all of them into account; and on the other hand, this allows to have an insight on what these algorithms are retrieving from the original data. This also motivates to ignore the individual statistical performances of these algorithms. Furthermore, looking at the topology, and the *prevalence* of topological structures of the output of such method, for such a geometric dataset (brain representations), seems to be a very interesting and promising idea.

**Weaknesses:**

Some techniques used to tackle this problem are not natural or are not motivated enough; either intuitively or theoretically, especially the prevalence-weighted wasserstein distance. In particular, I have the following comments:

 - The bijections between persistence diagrams usually add the points of the diagonal, with an infinite mass. In particular,
    - how is defined the prevalence on the diagonal?
    - If the diagonal has no mass here, there are also a few problems
        - A bijection may not exist (not the same cardinal)
        - This distance is not symmetric, as points of $d_2$ that are not matched to points of $d_1$ are not taken into account.
 - Multiplying points in the diagram by a real correspond to do an **homothety with center $(0,0)$**, which raises a few questions
    - To my knowledge, 0 has no particular role in a diagram, so what is the motivation for using it?
    - For a same prevalence $\alpha$, the rescaling depends at which scale the topological structure appears. And the same goes for the angle direction in which the points are moved. Is this behavior wanted, and why?

I think that taking into account the prevalence is an interesting idea, but I'm not convinced that this is the best way to do it.

Moreover, there are small typos:
 - Section 2.1
   - Feature types details could be improved, i.e., mathematical definitions.
 - Section 3.2.1
   - Mention who is $k$
   - The Vietoris Rips filtration is not a graph, but a clique complex (it has simplices of arbitrary dimensions)
   - This section could be made bigger, for non-TDA practitioners.
 - Section 2.3.2
   - Clarity could be improved here,
   - line 196 "multiple data element to [represent] the same homology [class]"
   - line 205 : intervals are never defined

**Questions:**

 1. Looking at the covariance-like matrices of the representations (feature-type) for each method, instead of looking directly at the topology they produce seems weird to me.
    - How to interpret the topology of these covariance matrices-like space?
    - How is this topological information relevant to inspect the  structures of the representations?
 2. The different features types are living on very different mathematical spaces,
    - How can you assert that your model is identifying brain representation structures rather than these mathematical spaces? These spaces may have significantly different scale densities, so this is not clear.
 3. The paper does intentionally not take into account pre-representation data.
    - Is the data fundamentally too large to be taken into account by TDA techniques?
    - How to identify if the feature extracted by the different representations are given by noise or useful structure from the dataset?
 4. The constants used to compute the prevalence are fixed at $R=1000$ bootstrap per resampling, at $90\%$, with homology of degree $1$.
    - How does these choices matter? i.e., do the results change with other parameters?
    - What is the motivation for using these values?

Overall, I think the proposed approach is currently too fuzzy and lacks proper motivation, which makes me lean towards rejection.

**Limitations:**

Limitations have been addressed.

---

> ### Author Rebuttal · Authors · 2023-08-10
>
> **Weaknesses**
>
> Some techniques used to tackle this problem are not natural or are not motivated enough [...], especially the prevalence-weighted wasserstein distance. In particular, I have the following comments:
> * The bijections between persistence diagrams usually add the points of the diagonal, with an infinite mass. In particular,
>     - how is defined the prevalence on the diagonal?
>         - Prevalence is defined only for nontrivial homology generators. It follows that prevalence is not defined on the diagonal, because no nontrivial generators of the persistence module are represented on the diagonal of the diagram.
>     - If the diagonal has no mass here, there are also a few problems
>         - A bijection may not exist (not the same cardinal)
>         - This distance is not symmetric, as points of d_2 that are not matched to points of d_1are not taken into account.
>             - These are both common practical issues in the use of the persistence diagram Wasserstein that are typically solved by the addition of a separate L^p penalty for added/removed points. We apologize for not more carefully distinguishing the definition of the Wasserstein distance on persistence diagrams from its definition on probability measures.
>
> * Multiplying points in the diagram by a real correspond to do an homothety with center (0,0), which raises a few questions [...]
>     * We are not multiplying points in the diagram by the prevalence value; rather, we are using the prevalence score to scale the contribution of a given generator to the p-Wasserstein cost. We apologize for the lack of clarity on this point in the manuscript.
>
> I think that taking into account the prevalence is an interesting idea, but I'm not convinced that this is the best way to do it.
>
> * We appreciate and understand your concern that the prevalence-weighted Wasserstein is untested and does not receive detailed theoretical treatment in this work. However, it does have precedent in other members of a family of weighted and generalized Wasserstein distances [1,2], though this family may generally fail to share some important properties of the unmodified Wasserstein distance [3]. Nonetheless, we feel it has a clear intuition: less-repeatable homology generators make a correspondingly lesser contribution to the Wasserstein transport cost between diagrams. We offer it as a first pass at combining both prevalence (statistical) and persistence (topological) information in a single measure and hope to characterize both its properties and alternatives in future work. In addition, we eagerly anticipate any refinements any other groups may propose to the combination of prevalence and persistence information.
>
> [1] T. de Wet, “Goodnes-of-fit tests for location and scale families based on a weighted L2-Wasserstein distance measure,” Test, vol. 11, no. 1, pp. 89–107, Jun. 2002, doi: 10.1007/BF02595731.
>
> [2] [1] B. Piccoli and F. Rossi, “Generalized Wasserstein distance and its application to transport equations with source,” Arch Rational Mech Anal, vol. 211, no. 1, pp. 335–358, Jan. 2014, doi: 10.1007/s00205-013-0669-x.
>
> [3] L. Lombardini and F. Rossi, “Obstructions to extension of Wasserstein distances for variable masses.” arXiv, Dec. 09, 2021. Accessed: Aug. 09, 2023. [Online]. Available: http://arxiv.org/abs/2112.04763
>
> **Questions**
> 1. [...]
>     - Covariance matrices have a well-characterized Riemannian structure (the symmetric positive-definite cone), which is why we use its geodesic distance as a choice of “dissimilarity measure” when comparing "covariance matrix-like" features (network matrices, spatial network matrices, and partial network matrices).
>     - The reviewer is correct to point out the relevance of the feature-space structure. It is quite possible that the "feature-produced" topological structure is the dominant (over parcellation & network definition) analytic choice influencing inter-subject variability -- indeed, this is precisely what we believe our results suggest. However, this is still useful information: for those working in the domain of application, it is important to know if and when feature choices overshadow DR choices.
> 2. Our paper does not make this assertion. In fact, one of our central interpretations of our results is that projections into feature space typically impose stronger topological constraints than do parcellation/network definition choices. However, we do claim that we find differences in representational structures. By comparing within feature type and across dimension reduction type, we can be confident that the topological differences we observe are due to the choice of brain representation.
> 3. [...]
>     * Yes. A full pre-representation dataset is on the order of ~10TB, which is larger than we are able to process with TDA and the resources we can access.
>     * This is where we find utility in the topological bootstrap; sampling stability is our best proxy distinction between signal and noise. However, it is not possible to be certain, because our data does not have a ground truth. See the “validation” section in our general response for further discussion.
> 4. Re: bootstrap params and homology degree
>     - Due to computational resource constraints, we were not able to fully characterize our results’ sensitivity to these parameters. However, intermediate testing work suggested that our results are similar (but with lower power) for R=100-250 bootstraps and resampling rate at 80%.
>     - We chose R = 1000 bootstraps to maximize the accuracy of our prevalence estimates.
>     - We chose a 90% resampling rate to keep the data structure relatively consistent while allowing a very large number of unique resamples.
>     - We restrict to H1 because (1) time and memory complexity for higher homology computations scales quickly and (2) minimal homology representatives are crucial to the topological bootstrap and can currently only be efficiently computed for H1.

---

> > ### Comment · Reviewer_SXyw · 2023-08-16
> > **Answer to rebuttal**
> >
> > Thank you for your explanations. I understand the point of the paper better now. However, I still think the lack of clarity, motivation and self-containedness, as well as lack of baseline experiments, are too salient to accept the work in its current form.

---

### Official Review · Reviewer_R8ML · 2023-07-09

**Soundness:** 4 excellent
**Presentation:** 4 excellent
**Contribution:** 4 excellent
**Rating:** 8
**Confidence:** 4

**Summary:**

A very interesting and rigorous exercise of comparison of embeddings for the purpose of evaluating manifold learning is presented. The chosen framework is root in trendy topological concepts such as persistent homology for the purposes of analyzing the data topology mixed with geometry-based measures of similarity across representations, and coupled with stochastic (topological) bootstrap to study variations over co-embeddings.
Contributions are explicitly mentioned in lns 113-117 and certainly delivered.
Best of my lot for this year.


**Strengths:**

+ The idea is exceptional IMHO. I’ve known of (and used myself) some other frameworks trying to establish and understand similarities or dissimilarities of projections, but they were all much more naïve. This one offers a clearly more sophisticated approach that yields a much richer picture without substantially sacrificing interpretability of results, with the “only” price to pay of a very large computational cost.
+ Extremely well explained despite the very complex concepts involved.
+ The observation on lns 290-1 that cycles with low persistence may carry meaningful structure is something that I have thought myself occasionally but couldn’t put my finger on it nor how to articulate it nor know how to reveal it. This is a very nice confirmation of that intuition.


**Weaknesses:**

Not many that I can see to be honest…
+ The study is only conducted in experimental data from the human connectome project but it is never validated in synthetic known manifolds with and without added noise. This means that some of the observations in the last part of the draft are (most likely correct but) difficult to verify. For instance, the implication that the proposed framework distinguishes more feature types and numbers than representation types.


**Questions:**

+ Lns 150-2: Is the separated treatment of the vectors and matrix intentional? Would a tensorial treatment lead to a more homogenous framework?
+ Lns 298-9: In order to “shift” the focus towards the representation, perhaps one can substitute Pearson’s correlations for Chatterjee’s correlations.


**Limitations:**

+ Lns 92-106 literature review is perhaps missing some of the most primitive approaches; e.g. distance distortions plots, (graph/manifold) isomorphisms, … this is possibly intentional as the persistence element there is only implicit rather than explicit as in the case of the works reviewed. But if it wasn’t, well, it is perhaps convenient to at least mention some early efforts.
+ Perhaps the number of compared embeddings (brain representations) is not as large as one would like for this type of exercise but the authors clearly state why they stay on low numbers (computational cost) and promise larger comparisons in the future. Looking forward to those!

---

> ### Author Rebuttal · Authors · 2023-08-10
>
> Questions:
>
> Lns 150-2: Is the separated treatment of the vectors and matrix intentional? Would a tensorial treatment lead to a more homogenous framework?
>
> - The treatment is intentional, since we chose dissimilarity metrics based on common methods of comparison in the application (neuroimaging) literature; these tend to use separate methods of comparison between the two, so we chose consistency with the literature over a homogeneous framework. In particular, tensorial treatments are typically reserved for matrix comparisons in the neuroimaging literature. However, a more homogeneous tensorial framework might yield more direct comparisons between methods; either choice of trade-off may yield helpful insights, and we appreciate the reviewer's thoughtful question.
>
> Lns 298-9: In order to “shift” the focus towards the representation, perhaps one can substitute Pearson’s correlations for Chatterjee’s correlations.
>
> - We thank the reviewer for this suggestion and will consider it for future iterations of this work.
>
>
> Limitations:
>
> Lns 92-106 literature review is perhaps missing some of the most primitive approaches…
>
> - We agree and apologize for this oversight.

---

> > ### Comment · Reviewer_R8ML · 2023-08-17
> >
> > I reckon mine was the easy answer as I truly like this work a lot. I can see some of the issues raised by other colleagues and while, fair enough, clarity for others may have not been so obvious (although it was to me) and self-containment could perhaps be improved, these are solvable issue after all. They do not seem to me to be inherent flaws of the method itself or invalidate the results, so I truly wish the best of luck to this submission after this rebuttal.

---

### Author Rebuttal · Authors · 2023-08-10

We thank the reviewers for their valuable input to our work. Please find our general response summary below, in addition to the item-wise responses we provide to individual reviewers. We also include figures showing typical persistence diagrams and a naive approach.

**Novelty and contribution**

Our primary goal in this work is to characterize the impact of analytical flexibility in neuroimaging analysis pipelines. Analytical flexibility is a major challenge in psychology and neuroimaging that encompasses the wide range of acceptable analysis steps and parameter decisions available to researchers. Analytical flexibility leads to tens of thousands of different versions of valid analysis pipelines with extensive variability in results. Importantly, analytical flexibility has a siloing effect on the field because it restricts cross-pollination of findings and ideas to occur only within analytically aligned subsets of the scientific community. Analytical flexibility is especially challenging to characterize and address in resting state neuroimaging because feature sets differ in both size (depending on parcellation/network definition) and structure (depending on the feature choice).

The major novel contribution of this work is to develop a mathematical framework that enables robust and generalizable statistical comparisons between resting state neuroimaging features sets that differ in size and nature. Importantly, the findings reveal that differences between neuroimaging feature sets are primarily driven by the feature choice rather than by parcellation or network definition. These results suggest that future work into clinical and behavioral neuroimaging correlates should focus more on feature type comparisons and less on parcellation choices.

Separately, we also believe this work makes an important novel contribution in its use of the topological bootstrap to show “that cycles with low persistence may carry meaningful structure” in high-dimensional real data with complex topology.

**Terminology**

Our reviewers have collectively pointed out some ambiguities in our terminology around “dimension reduction,” “brain representations,” and “manifold learning”. We agree with these comments and apologize for our lack of clarity. The goal of this study was to compare different feature sets derived from resting state neuroimaging data. Reviewer 5 correctly notes that the extraction of these feature sets typically involves a spatial summary of the brain into parcels or voxel-weighted networks, each of which is characterized by timeseries. From these timeseries, a variety of features can be calculated including amplitudes (overall strength), functional connectivity (between-parcel temporal correlations), etc. We agree that the terminology of ‘parcellation’ and ‘feature set’ is preferable over dimension reduction (which is too domain-general), brain representation (which encompasses both parcellation and feature choice), and manifold learning (which we use primarily to refer to parcellation). We did not investigate more general dimensionality reduction approaches such as t-SNE and UMAP because these techniques are relatively uncommon in the neuroimaging domain. Nevertheless, we believe that our flexible analysis framework extends to broader applications beyond the domain-specific comparisons performed here.

**Relevance to NeurIPS**

Though our motivation is firmly rooted in the neuroimaging domain (as explained above), we also believe our investigation offers meaningful contributions more broadly.

First, to our knowledge, most of the literature evaluating dimension reduction considers the regime where (a) the ambient dimension is smaller than the number of data samples and (b) the target dimension is low (often two or three dimensions). By contrast, we work in an ill-conditioned (samples << original dimension) regime with high original and target dimension, which is common in many DR use cases. As such, our domain-specific analyses offer a valuable testbed for very high-dimensional data.

Secondly, we believe that our method offers an extremely broad and flexible framework for future comparisons of dimension reduction beyond the field of neuroimaging. In particular, because we can conduct this analysis on any datasets compatible with Vietoris-Rips filtration, we are not limited to, e.g., comparing graphs with identical vertex sets, and can consider broad classes of data and methods.

**Motivation for validation approach**

Understandably, several reviewers commented on our decision not to validate our analysis in synthetic manifolds or simulated data, and, relatedly, our method’s ability to distinguish useful structure and noise. We apologize for not making our reasoning clearer in the manuscript. While we agree that manipulable validation is very imortant, we do not feel that simulations or synthetic manifolds are viable options in our problem context. Because brain activity simulation remains an open problem and synthetic manifolds tend to have different structure, sampling, and dimensionality characteristics than our data, we validated via statistical stability instead of synthetic data.

Though we do not currently have plans for synthetic/simulation testing of this method, we are planning to characterize topological behavior of several relevant null models in future work; because random geometric complexes remain under active theoretical investigation ([1-3]), the empirical characterization of null models in our data regime is of interest.

[1] D. Yogeshwaran, E. Subag, and R. J. Adler, “Random geometric complexes in the thermodynamic regime.” arXiv, Sep. 09, 2015. doi: 10.48550/arXiv.1403.1164

[2] O. Bobrowski, M. Kahle, and P. Skraba, “Maximally Persistent Cycles in Random Geometric Complexes.” arXiv, May 15, 2016. doi: 10.48550/arXiv.1509.04347

[3] O. Bobrowski and M. Kahle, “Topology of random geometric complexes: a survey.” arXiv, Jul. 23, 2017. doi: 10.48550/arXiv.1409.4734

---

### Decision · Program_Chairs · 2023-09-21

**Decision:**

Reject

**Comment:**

Reviewers mostly agreed on the relevance of the subject tackled in this paper but were somewhat split concerning their overall recommendation. It is clear to me and the reviewers that this paper addresses an important topic, but numerous concerns were raised from experts in topology-driven analysis:

1. A lack of clarity and a presentation that is not entirely self-contained.
2. A lack of (baseline) experiments.
3. A lack of theoretical justifications for some of the choices (in particular the prevalence-weighted Wasserstein distance was mentioned as an example here; as an expert in the subject matter myself, I would also appreciate a more in-depth discussion of this).

While some of the issues could be addressed during the rebuttal, the lack of baseline experiments and the overall presentation issues are still present. Were this a journal paper, I would suggest to resubmit with major revisions, but within the conference cycle, another round of reviews is unfortunately not possible. While I thus cannot endorse this paper for publication, I encourage the authors to use the feedback to improve the paper and resubmit it to a machine learning conference. The work is definitely in scope and showcases the potential of a topology-driven analysis in neuroscience.

Along these lines, the additional works by reviewer `MybT` might be helpful to study as related work from a topological representation learning perspective:

- Barannikov et al., "Representation Topology Divergence: A method for Comparing Neural Network Representations"
- Moor et al., "Topological Autoencoders"
- Trofimov et al,, "Learning Topology-Preserving Data Representations"

On a more general level, the paper might benefit from a discussion of the overall utility of a topology-based perspective here. To this end, the following works might prove helpful:

- Sizemore et al., "The Importance of the Whole: Topological Data Analysis for the Network Neuroscientist"
- Rieck et al., "Uncovering the Topology of Time-Varying fMRI Data using Cubical Persistence"
- Stolz, "Computational Topology in Neuroscience"

I understand that this is not the desired outcome for the authors but hope that they find the reviews as constructive as I did. The decision to suggest rejection for this work was not taken lightly, and I look very much forward to seeing this work published in some future venue—I believe it has the potential to make a strong impact!